# Chamigrane-Type Sesquiterpenes from *Laurencia dendroidea* as Lead Compounds against *Naegleria fowleri*

**DOI:** 10.3390/md21040224

**Published:** 2023-03-31

**Authors:** Iñigo Arberas-Jiménez, Nathália Nocchi, Javier Chao-Pellicer, Ines Sifaoui, Angélica Ribeiro Soares, Ana R. Díaz-Marrero, José J. Fernández, José E. Piñero, Jacob Lorenzo-Morales

**Affiliations:** 1Instituto Universitario de Enfermedades Tropicales y Salud Pública de Canarias, Universidad de La Laguna (ULL), Avda. Astrofísico Fco. Sánchez, S/N, 38206 La Laguna, Tenerife, Islas Canarias, Spainjpinero@ull.edu.es (J.E.P.); 2Departamento de Obstetricia y Ginecología, Pediatría, Medicina Preventiva y Salud Pública, Toxicología, Medicina Legal y Forense y Parasitología, Universidad de La Laguna (ULL), 38206 La Laguna, Tenerife, Islas Canarias, Spain; 3Instituto Universitario de Bio-Orgánica Antonio González (IUBO AG), Universidad de La Laguna (ULL), Avda. Astrofísico Fco. Sánchez 2, 38206 La Laguna, Tenerife, Islas Canarias, Spain; 4Departamento de Química Orgánica, Universidad de La Laguna (ULL), Avda. Astrofísico Fco. Sánchez 3, 38206 La Laguna, Tenerife, Islas Canarias, Spain; 5Consorcio Centro de Investigación Biomédica En Red (CIBER) de Enfermedades Infecciosas (CIBERINFEC), Instituto de Salud Carlos III, 28006 Madrid, Spain; 6Instituto de Biodiversidade e Sustentabilidade, Universidade Federal do Rio de Janiero, Avda. São José do Barreto, 764, Macaé 27965-045, RJ, Brazil; 7Instituto de Productos Naturales y Agrobiología (IPNA), Consejo Superior de Investigaciones Científicas (CSIC), Avda. Astrofísico Fco. Sánchez 3, 38206 La Laguna, Tenerife, Islas Canarias, Spain

**Keywords:** elatol, marine chamigrane, sesquiterpenes, *Naegleria fowleri*, *Laurencia dendroidea*

## Abstract

*Naegleria fowleri* is an opportunistic protozoon that can be found in warm water bodies. It is the causative agent of the primary amoebic meningoencephalitis. Focused on our interest to develop promising lead structures for the development of antiparasitic agents, this study was aimed at identifying new anti-*Naegleria* marine natural products from a collection of chamigrane-type sesquiterpenes with structural variety in the levels of saturation, halogenation and oxygenation isolated from *Laurencia dendroidea*. (+)-Elatol (**1**) was the most active compound against *Naegleria fowleri* trophozoites with IC_50_ values of 1.08 μM against the ATCC 30808™ strain and 1.14 μM against the ATCC 30215™ strain. Furthermore, the activity of (+)-elatol (**1**) against the resistant stage of *N. fowleri* was also assessed, showing great cysticidal properties with a very similar IC_50_ value (1.14 µM) to the one obtained for the trophozoite stage. Moreover, at low concentrations (+)-elatol (**1**) showed no toxic effect towards murine macrophages and could induce the appearance of different cellular events related to the programmed cell death, such as an increase of the plasma membrane permeability, reactive oxygen species overproduction, mitochondrial malfunction or chromatin condensation. Its enantiomer (−)-elatol (**2**) was shown to be 34-fold less potent with an IC_50_ of 36.77 μM and 38.03 μM. An analysis of the structure–activity relationship suggests that dehalogenation leads to a significant decrease of activity. The lipophilic character of these compounds is an essential property to cross the blood-brain barrier, therefore they represent interesting chemical scaffolds to develop new drugs.

## 1. Introduction

Free-living amoebae are opportunistic protozoa that can complete their life cycle without the need to infect a host [1] and inhabit any type of environment. Predominantly found in water and soil, they can also use the air to disperse [2]. These organisms are defined as unicellular living beings endowed with motility (amoeboid motility) through a series of cytoplasmic prolongations called pseudopodia and able to feed by phagocytosis [3].

*Naegleria fowleri* is the only free-living amoeba of the *Naegleria* genus capable of infecting the human [4], and causes a fulminant disease affecting the central nervous system called primary amoebic meningoencephalitis (PAM), with a case fatality rate of over 95%. This amoeba has three stages throughout its life cycle: a trophozoite form, which is the infective stage and through which they feed and reproduce. This trophozoite can transform to a cyst stage, when environmental conditions are not suitable, such as high temperatures, changes in environmental pH or high salinity [5] or, in turn, to a flagellar form when there is an ionic change in the medium [6].

This parasite causes infections, in most cases, in children or young people whose immune system is suppressed or immature [7]. Most infections begin by the inhalation of contaminated water containing the amoeba in the trophozoite form. Once they enter through the nasal mucosa, the trophozoites bind to the olfactory bulb and olfactory mucosa, as the cells express G-protein-coupled receptors (GPCR) NF0059410, a structural homologue of the human M1 muscarinic receptor. Upon the binding of acetycholine and the GPCR, chemotaxis is established, facilitating the movement of the amoeba into the brain through the cribriform plate [8], where it will begin to destroy the brain tissue by the food cups and the release of cytolytic molecules [7,9].

Regarding the symptoms, these are characterized by a clinical picture similar to the meningitis caused by bacteria or viruses. These signs and symptoms usually appear during the first few days after infection and usually include fever, headache, vomiting, neck stiffness and convulsions. A few days after the onset of these first sign and symptoms, the intracranial and cerebrospinal fluid pressure increases preceding the death of the patient [7,10].

As to the diagnostic, once the sample has been extracted from the lumbar puncture, three main methods have to be performed together for the detection of the parasite: the morphological analysis of the micro-organisms from a wet cerebrospinal fluid preparation under a microscope [11], performing differentiation tests by subjecting the parasite to further stages and molecular identification by a PCR confirming the previous morphological findings [12,13].

Furthermore, despite being a fulminant disease with a high fatality rate, there is no standardized treatment against it. Currently, the therapeutic guideline consists of a combined action of amphotericin B and miltefosine, whose antiprotozoal activity has been demonstrated in other parasites [14,15], together with azoles, where the synergy of all these compounds is necessary to avoid parasite resistance. These drugs are administered intravenously and in some cases by intrathecal injection. Miltefosine, together with azithromycin, is one of the few drugs that is administered orally [12,16]. In addition, in some cases, the patient is induced into a state of hypothermia in order to stop the increase of the intracranial pressure and the progression of the amoeba. Despite treatment, only a very small percentage of patients survive, and many of them suffer irreparable damage after treatment with the drugs [17]. However, the current treatment options involve a wide range of severe side effects. For this reason, it is essential to search for novel products that demonstrate high selectivity against the parasite. 

The natural and marine environments are a potential source of structurally diverse and biologically active secondary metabolites that offer the opportunity to design antiprotozoal drugs [18,19,20]. Particularly, the marine macroalgae are considered as a promising source of novel biochemically active compounds against protozoa. These organisms have developed different strategies to defend themselves from the environmental risks that have resulted in a significant level of compounds with high biotechnological or pharmaceutical applications [21]. Among all marine macroalgae, the red algae of the genus *Laurencia* are accepted unanimously as a prolific producer of secondary metabolites, highlighting the sesquiterpenes, diterpenes, triterpenes and C15 acetogenins [22,23]. Many compounds of *Laurencia* exhibit important antiparasitic properties against several parasites [24,25,26,27,28,29].

The sesquiterpenes isolated from the genus *Laurencia* currently constitute more than 575 structures reported. Among them, the chamigrane-type sesquiterpenes are the most frequently occurring class, accounting for approximately 30% of all isolated sesquiterpenes. The chamigranes are characterized by presenting a spiro [5.5]undecane skeleton in which a stereogenic quaternary carbon (C-6) joins the spirocycles, with much variability of substitution with functional groups. This compound class has attracted great interest from the structural and pharmacological point of view, due to the number of skeletal types [23,30,31].

Focused on our interest to develop promising lead structures for the development of antiparasitic agents, this study aimed to identify new anti-*Naegleria* marine natural products from a collection of chamigrane-type sesquiterpenes isolated from *Laurencia dendroidea* (Figure 1).

## 2. Results

### 2.1. Chamigrane-Type Sesquiterpenes from Laurencia dendroidea

A collection of ten known chamigrane-type sesquiterpenes (**1**–**10**) (Figure 1) with structural variety in the level of saturation, halogenation and oxygenation were isolated from the organic extracts of specimens of the *Laurencia dendroidea* collected in distinct areas of the southeastern Brazilian coast: (+)-elatol (**1**); (−)-elatol (**2**); (+)-debromoelatol (**3**); (+)-dechloroelatol (**4**); (−)-rogiolol (**5**); (−)-3,10-dibromo-4-chloro-alpha-chamigrane (**6**); (−)-dendroidiol (**7**); (−)-cartilagenol (**8**); (+)-obtusol (**9**); and (+)-obtusane (**10**) [32,33].

The compounds **1** and **2** were identified as the enantiomers (+)-elatol (**1**) and (−)-elatol (**2**). These metabolites showed identical NMR and mass spectrometry (MS) experimental data (see Appendix A). However, they exhibit opposite optical rotations and equal magnitude: ${\left[\alpha \right]}_{D}^{20}$ +80 (*c* 0.20, CHCl_3_) and ${\left[\alpha \right]}_{D}^{20}$ −80 (*c* 0.20, CHCl_3_) for compounds **1** and **2**, respectively. These data confirm that it is the enantiomers dextrorotatory or (+) and levorotatory or (–) of elatol (**2**).

### 2.2. In Vitro Activity against Naegleria fowleri Trophozoites and Cytotoxicity Assays

The in vitro activity tests demonstrated that (+)-elatol (**1**), (−)-elatol (**2**), and (−)-rogiolol (**5**) were active against *N. fowleri* in both ATCC 30808™ and ATCC 30215™ strains at lower concentrations than the reference drug miltefosine. Additionally, (+)-elatol was active at similar concentration values as amphotericin B. Moreover, these compounds were not toxic at low concentrations. All these results are summarized in Table 1.

### 2.3. In Vitro Activity against Naegleria fowleri Cysts

Furthermore, the cysticidal activity of (+)-elatol (**1**) was also assessed against the ATCC 30808™ strain. The results showed an IC_50_ value of 1.14 ± 0.09 µM against the resistant phase of the amoeba, which is practically identical to the values obtained for the trophozoite stage.

The high activity of (+)-elatol (**1**) against both the trophozoite and cyst stages and its specificity in the action (low cytotoxicity) lead us to continue with further experiments in order to analyze the biological and chemical changes that this compound induces in treated amoebae.

### 2.4. Programmed Cell Death (PCD) Induction Evaluation

The assessment of the type of cell death induced by (+)-elatol was conducted using several techniques to highlight the different cellular events considered as PCD markers.

These experiments were carried out by using the ATCC 30808™ type strain and the IC_90_ value (2.45 µM) of the (+)-elatol (**1**).

#### 2.4.1. (+)-Elatol (**1**) Induces Chromatin Condensation

The presence of condensed chromatin is one of the most characteristic metabolic events that occurs during the PCD [34]. In this assay, a double-stain apoptosis detection kit (Hoechst 33342/Propidium Iodide (PI)) was used. Figure 2 and Appendix A show an intense blue fluorescence in (+)-elatol (**1**)-treated cells, whereas no sign of that fluorescence can be observed in negative control amoebae. Moreover, the absence of the red fluorescence that corresponds to PI suggests that these cells were undergoing an early apoptotic phase.

#### 2.4.2. Plasma Membrane Permeability

The plasma membrane permeability was also evaluated using the SYTOX™ Green stain. This stain is impermeable to cells in healthy conditions; however, it crosses the plasma membrane when it is damaged and binds the DNA increasing its fluorescence 500 times. As it can be seen in Figure 3 and Appendix A, 24 h treated amoeba showed a significantly (*p* < 0.0001 (****)) higher green fluorescence compared to the negative control. Nevertheless, the integrity of the cell membrane was maintained keeping the cellular content inside. In fact, no fluorescence can be observed in the media, hence no genetic material could be found outside the cells.

#### 2.4.3. Reactive Oxygen Species Production

The generation of reactive oxygen species (ROS) was also measured with the CellROX^®^ Deep Red assay kit. As shown in Figure 4 and Appendix A, (+)-elatol (**1**) induces the over-generation of ROS after 24 h of incubation with the IC_90_ of the compound. The percentage of stained cells after the treatment is significative relative to the negative control, *p* < 0.0001 (****).

#### 2.4.4. Analysis of Disorders in the Mitochondrial Function

Two different assays were carried out to determine the mitochondrial damage provoked by (+)-elatol (**1**). On the one hand, the collapse of the mitochondrial membrane potential was evaluated using the JC-1 dye (Cayman Chemical). Negative control cells showed an intense red fluorescence emitted by the J-aggregates of the JC-1 stain that remained in the mitochondria in healthy conditions. However, when the amoebae were treated with the IC_90_ of (+)-elatol (**1**), a depolymerization of the mitochondrial membrane potential, shown as the green fluorescence of the JC-1 monomers, could be observed (Figure 5 and Appendix A). Additionally, the ratio between the mean fluorescence intensity emitted by the aggregate form and the monomer form showed a significant difference between the negative control and the treated cells; *p* < 0.001 (***).

The ATP production was also measured with the aim to confirm the mitochondrial damage. The treatment with (+)-elatol (**1**) reduced the ATP production in 82.11% compared to the untreated cells (Figure 6).

## 3. Discussion

In this study, the anti-*Naegleria* activity of 10 chamigrane-type sesquiterpenes obtained from the red algae *L. dendroidea* was evaluated. Among the tested compounds, (+)-elatol (**1**) was the most active one against *Naegleria fowleri* trophozoites with IC_50_ values of 1.08 μM against the ATCC 30808™ strain and 1.14 μM against the ATCC 30215™ strain (Table 1). Furthermore, the activity of (+)-elatol (**1**) against the resistant stage of *N. fowleri* was also assessed, showing great cysticidal properties with a very similar IC_50_ value (1.14 µM) to the one obtained for the trophozoite stage.

An interesting aspect in this study from the structural and biological perspective is the fact that a comparison with the enantiomer of the most active molecule was possible. Thus, (−)-elatol (**2**) was shown to be 34-fold less potent with an IC_50_ of 36.77 μM and 38.03 μM, than (+)-elatol (**1**) (Figure 7). Additionally, the loss of any halogen atoms, such as bromine at C-10 or chlorine at C-2, leads to a significant decrease of activity ((+)-debromoelatol (**3**), IC_50_ > 98.11 μM and (+)-dechloroelatol (**3**), IC_50_ > 83.54 μM).

On the other hand, (−)-rogiolol (**5**), which shares the same fragment C-6-C-11 as compound 1 and a bromine atom at C-2, showed an intermediate activity value (IC_50_ 22.50 µM and IC_50_ 9.71 µM against *N. fowleri* strains ATCC 30808™ and ATCC 30215™, respectively). The spatial disposition of functional groups of compounds 1 and 5 compared with that of compound 2 could be a determinant for the biological activity as showed in Figure 8.

These results against both stages of *N. fowleri* lead us to continue with further assays in order to elucidate the mechanism of action of (+)- elatol (**1**). In 2017, the PCD process in the *Naegleria* genus was firstly characterized by Cárdenas-Zuñiga et al. [35] and the typical markers to describe it were reported. The treatment of *N. fowleri* trophozoites with (+)-elatol (**1**) showed the presence of different metabolic events that are characteristic of the PCD process such as chromatin condensation, permeability of the plasma membrane, increase of the ROS production or mitochondrial damage.

The development of novel treatments against the “brain eating” amoeba is not only based on the search of active compounds with low cytotoxicity, but also the induction of the PCD process is desired. During the necrotic pathway, cytosolic constituents are poured into the intercellular space provoking the inflammation of the surrounding tissues and the appearance of side effects. However, through the PCD, the inflammatory process is avoided leading to a safer therapy [36,37].

Moreover, the blood-brain barrier constitutes a very important limiting factor when speaking about new anti-*Naegleria* compounds. In fact, the poor penetration of Amphotericin B (one of the reference drugs against the PAM) to the central nervous system is one of the reasons behind the poor clinical outcomes of this disease [38]. (+)-elatol (1) meets some of the requirements for crossing the blood-brain barrier [39] such as the low molecular weight (333.69 Da) and high lipophilicity with a calculated log P of 3.65 [40].

This work represents a first approach to elucidate the in vitro activity and the mechanism of action of (+)-elatol against *N. fowleri* using fluorescence microscopy. A proteomic study could deepen and confirm the effect of the bioactive molecule in the present protozoa.

## 4. Materials and Methods

### 4.1. Biological Material

Specimens of *Laurencia dendroidea* were collected from the intertidal zone of distinct areas of the southeastern Brazilian coast (20°11′13.9″ S, 040°11′25.4″ W to 23°01′33.7″ S, 044°14′08.1″ W) [32,33]. Immediately after collection, the samples were deposited in polyethylene bags with a hermetic lock, frozen and transported to the laboratory. Voucher specimens were deposited at the Herbarium of the Rio de Janeiro Federal University, Brazil (RFA). Voucher specimens are deposited under codes: RFA 36068–Biscaia Inlet–Angra dos Reis–Rio de Janeiro state; RFA 35887–Manguinhos Beach–Serra–Espírito Santo state; RFA 36045–Vermelha Beach–Parati–Rio de Janeiro state; RFA 38846–Azeda Beach–Armação dos Búzios–Rio de Janeiro state.

The frozen specimens of *Laurencia dendroidea* were freeze-dried. From the lyophilized samples, different extracts were obtained in EtOAc/MeOH (1:1) by maceration at room temperature assisted by ultrasound for 15 min, followed by filtration and concentration in a rotary evaporator under reduced pressure at 40 °C. The extracts were initially submitted to size exclusion chromatography using Sephadex LH-20 (6 × 34 cm, Pharmacia Fine Chemicals^®^) and eluted with n-hexane/CH_2_Cl_2_/MeOH (2:1:1). The chamigrane-enriched fractions were processed by medium-pressure with column Lobar^®^ Gröbe B (310-25) LiChroprep^®^ Si-60 (40–63 μm, MERCK^®^) followed by high-performance liquid chromatography (HPLC) (Agilent 1260 Infinity Quaternary LC equipped with a Diode Array Detector (Waldbronn, Germany)) using a μ-Porasil™ silica column (125 Å; 1.9 × 15 cm) and n-hexane/EtOAc mixtures of increasing polarity adapted to the sample nature. The structures of the compounds were determined by spectroscopic and spectrometric analysis (1D and 2D NMR, HR-ESI-MS and ${\left[\alpha \right]}_{D}^{20}$), as well as by comparison with reported data. 1D and 2D nuclear magnetic resonance (NMR) spectra were recorded on a Bruker AVANCE 500 MHz or 600 MHz equipped with a 5 mm TCI inverse detection cryo-probe (Bruker Biospin, Falländen, Switzerland) when required. The ^1^H and ^13^C NMR chemical shifts were reported in ppm and referenced to internal residual solvent deuterochloroform (CDCl_3_) (99.9%) at 300 K (δ_H_ 7.26 ppm; δ_C_ 77.0 ppm). The NMR experiments were performed using standard pulse sequences and processed using the MestReNova software (v.10., Santiago de Compostela, Spain).

### 4.2. Cell and Culture Maintenance

The in vitro experiments were carried out using two different *Naegleria fowleri* type strains, ATCC 30808™ and ATCC 30215™. The cells were grown in axenic conditions as previously described in other works of our group [29]. The strains were cultured in a level 3 biological security facility at the Instituto Universitario de Enfermedades Tropicales y Salud Pública de Canarias, Universidad de La Laguna, Spain.

The cytotoxicity assays were carried out in a murine macrophages cell line (ATCC TIB-67). The cells were incubated in Dulbecco’s Modified Eagle’s medium (GIBCO, Thermo Fisher Scientific, Madrid, Spain) (DMEM, *w/v*) supplemented with 10% (*v/v*) fetal bovine serum (FBS) and 10 µg/mL of gentamicin (Sigma-Aldrich, Madrid, Spain) at 37 °C in a 5% of CO_2_ atmosphere.

For the in vitro cysticidal activity assays, the *Naegleria fowleri* ATCC 30808™ strain was used. Cysts were obtained transferring the bactocasitone grown trophozoites to the MYAS medium and keeping them for ten days in slight agitation [41].

### 4.3. In Vitro Activity Evaluation against Naegleria fowleri Trophozoites

An alamarBlue™-based colorimetric assay was used to determine the in vitro activity of the chamigrane-type sesquiterpenes. Briefly, a known concentration of the trophozoites (2 × 10^5^ cells/mL) was incubated with serial dilutions of the compounds. After, the alamarBlue™ reagent was added and the plate was placed at 37 °C for 48 h. Finally, the fluorescence was read and a non-linear regression analysis with a 95% confidence limit, in order to reckon the 50% and 90% inhibitory concentration (IC_50_ and IC_90_), was performed [28].

### 4.4. In Vitro Cytotoxicity Determination

The in vitro cytotoxicity of the compounds was determined following the alamarBlue ™ protocol mentioned in the previous section. Murine macrophages were incubated with serial dilutions of the evaluated molecules for 24 h at 37 °C and in presence of 5% of CO_2_. Finally, the 50% cytotoxic concentration (CC_50_) was determined after reading the fluorescence.

### 4.5. In Vitro Cysticidal Activity Evaluation

The in vitro activity of the compounds against the *Naegleria fowleri* cyst was determined using an alamarBlue™-based fluorometric assay as previously described [41]. Mature cysts were incubated with serial dilutions of the evaluated compound for 24 h at 37 °C. After that, the medium was removed and fresh bactocasitone was added. Later, the alamarBlue^®^ reagent was placed and the plates were incubated for 72 additional hours at 37 °C. Finally, the EnSpire Multimode Plate Reader (PerkinElmer, Madrid, Spain) was used to read the fluorescence and calculate the IC_50_ as described in Section 4.3.

### 4.6. Evaluation of Programmed Cell Death Induction

For the determination of the type of cell death that induces (+)-elatol (**1**), different commercial kits were used. To carry out the assays, *Naegleria fowleri* ATCC 30808™ trophozoites were previously incubated with the IC_90_ (2.45 µM) of (+)-elatol (**1**) for 24 h. Next, the reagents corresponding to each kit were added, allowing the appearance of programmed cell death (PCD) processes in the cells to be observed. Moreover, the percentage of stained cells was determined using the EVOS™ M5000 Software (Invitrogen by Thermo Fisher Scientific). All experiments were performed in triplicate and five different images (×40) were processed each time.

#### 4.6.1. Chromatin Condensation

The condensed chromatin was detected with the double-stain apoptosis detection kit (Hoechst 33342/PI) (Life Technologies, Madrid, Spain). The Hoechst 33342 stain emits blue fluorescence in the presence of condensed chromatin while the Propidium iodide (PI) is able to travel through the membrane of dead cells and emit a red fluorescence. Hence, healthy cells will not show any red fluoresce nor intense blue fluorescence.

#### 4.6.2. Evaluation of Plasma Membrane Permeability

The permeability of the plasma membrane was assessed with the SYTOX Green^®^ stain. The plasma membrane is impermeable to the dye in healthy conditions. However, when the permeability of the plasma membrane is compromised, the SYTOX Green^®^ enters the cells, binds the DNA and emits an intense green fluorescence at 523 nm wavelength.

#### 4.6.3. Oxidative Stress Induction

To determine the oxidative damage, the CellROX Deep Red fluorescent assay (Thermo Fisher Scientific) was used. For this assay, *Naegleria fowleri* trophozites were incubated with (+)-elatol (**1**) for 24 h, before the addition of the CellROX dye. These reagents produce red fluorescence with maximum absorption/emission peaks of ~644/665 nm when facing the reactive oxygen species (ROS) in the cytoplasm.

#### 4.6.4. Mitochondrial Function Disruption

For determining mitochondrial damage in treated amoebae, two different assays were performed. Firstly, to measure the collapse of the transmembrane electrochemical gradient, a lipophilic cationic dye called JC-1 (Cayman Chemicals Vitro SA, Madrid, Spain) was used. This compound selectively enters the mitochondria and is able to change the color reversibly from red to green as the membrane potential decreases (ΔΨm). When the membrane potential is high, the fluorophore will be found in the mitochondria in the j-aggregates form emitting red fluorescence at 595 nm. On the other hand, when membrane depolarization occurs, the reagent disperses and remains in its monomeric form showing green fluorescence at 535 nm.

Afterwards, to validate the mitochondrial damage, the levels of ATP produced in the amoebae treated with the IC_90_ of the compound were analyzed using the CellTiter-Glo^®^ Luminescent Cell Viability Assay reagent (Promega Biotech Ibérica S.L, Madrid, Spain) following manufacturer’s instructions.

### 4.7. Statistical Analysis

The in vitro IC_50_, IC_90_ and CC_50_ were determined by a non-linear regression analysis with a 95% of confidence limit using the Graphpad Prism version 9.0 (GraphPad Software; CA; USA). Three independent experiments were performed to obtain each data, which is expressed as mean ± standard deviation (SD).

The percentage of stained cells determination in the PCD assays was calculated using the EVOS™ M5000 Software (Invitrogen by Thermo Fisher Scientific). For each kit, five different images were processed with a minimum of 80-100 cells per image. Bar graphs represent the mean percentage of stained cells and the SD of both treated and untreated amoebae. The bar graph of the mitochondrial membrane potential evaluation illustrates the mean fluorescence and SD emitted by the aggregate and monomer forms of the JC-1 stain. Three different experiments were performed for each kit. Moreover, in all the PCD assays, a one-way analysis of variance (ANOVA) was assessed to study the differences between the treated and negative control trophozoites. ** *p* < 0.01; *** *p* < 0.001; **** *p* < 0.0001 significant differences.

## 5. Conclusions

In this study, the activity against two strains of *Naegleria fowleri* of a collection of chamigrane-type sesquiterpenes with structural variety in the level of saturation, halogenation and oxygenation isolated from *Laurencia dendroidea* was evaluated. Among all tested compounds, (+)-elatol (**1**) was the most active compound against trophozoites of *Naegleria fowleri*, strains ATCC 30808™ and ATCC 30215™, with IC_50_ values of 1.08 μM and 1.14 μM, respectively. The activity of (+)-elatol (**1**) against the resistant stage of *N. fowleri* was also assessed, showing great cysticidal properties with a very similar IC_50_ value (1.14 µM) to the one obtained for the trophozoite stage. Its enantiomer (−)-elatol (**2**) was also analyzed, showing to be 34-fold less potent with an IC_50_ of 36.77 μM and 38.03 μM, thus revealing that spatial disposition is crucial for the anti-*Naegleria* activity. Additionally, the structure–activity relationship (SAR) analysis suggests that dehalogenation leads to a significant decrease of activity. The lipophilic character of these compounds, such as (+)-elatol (**1**) (log P 3.65), confers them as good chemical scaffolds to develop new drugs to cross the blood-brain barrier.

## 6. Patents

Protected under this number of registration: P202230496

## Figures and Tables

**Figure 1 marinedrugs-21-00224-f001:**
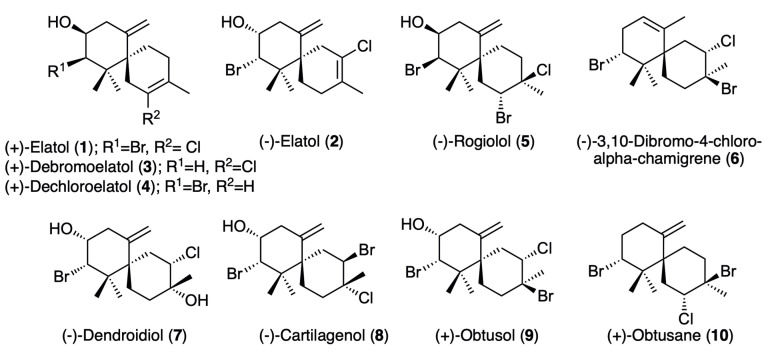
Chamigrane-type sesquiterpenes isolated from *Laurencia dendroidea*.

**Figure 2 marinedrugs-21-00224-f002:**
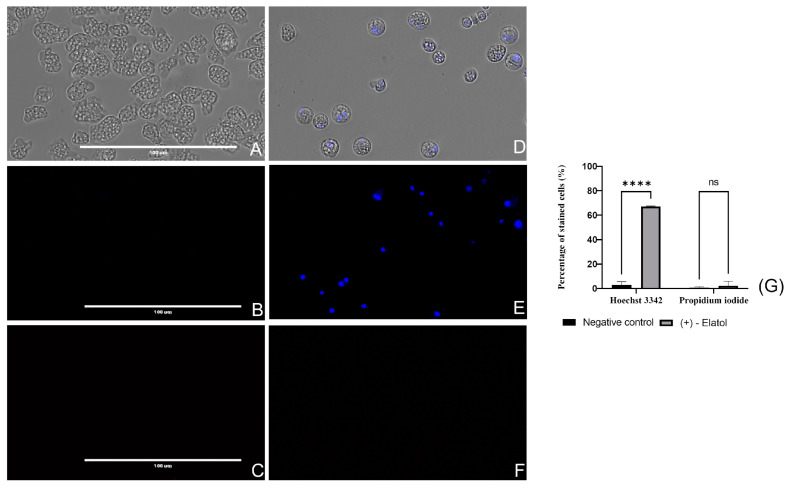
*Naegleria fowleri* trophozoites incubated with IC_90_ of (+)-elatol (**1**) for 24 h (**D**–**F**), negative control (**A**–**C**). Condensed chromatin can be seen in treated cells (**E**) since the Hoechst stain emits an intense fluorescence while no blue stained cells can be observed in the negative control cells (**B**). The iodide channel (**C**,**F**) shows no red fluorescence in the cells incubated with the yucatecone (**F**) nor in the non-treated cells (**C**), meaning that the PI stain was not permeable to the cells. Images (×40) are representative of the observed cell population and were obtained in the EVOS M500 Cell Imaging System, Life Technologies, Spain. Scale bar: 100 µm. (**G**) The bar graph represents the mean percentage of the stained cells and the standard deviation (SD) after the performance of three different assays. Differences between the values were assessed using a one-way analysis of variance (ANOVA); **** *p* < 0.0001; ns = no significant. The measurement was performed in the EVOS™ M5000 Software (Invitrogen by Thermo Fisher Scientific). Five different images were evaluated each time.

**Figure 3 marinedrugs-21-00224-f003:**
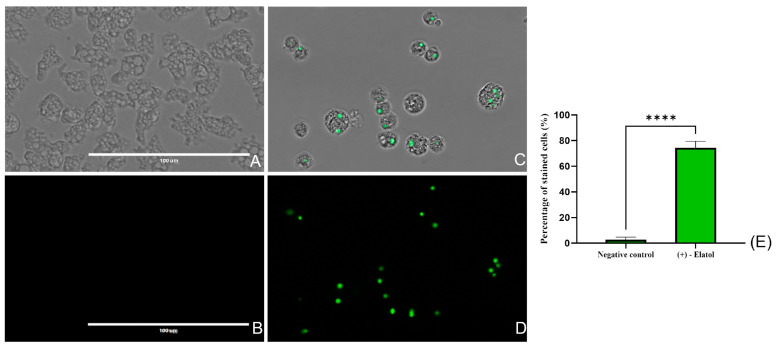
Evaluation of plasma membrane integrity of *Naegleria fowleri* (ATCC 30808™) trophozoites after the treatment with (+)-elatol (**1**) for 24 h. For this assay, the SYTOX^®^ green dye was used. The stain is impermeable to healthy (untreated) cells in which fluorescence is not observed (**A**,**B**). However, the plasma membrane is damaged in treated cells, allowing the SYTOX^®^ green to travel through it and bind to the DNA emitting an intense green fluorescence (**C**,**D**). Overlay channel (**A**,**C**) and GFP channel (**B**,**D**). Images (×40) were obtained using an EVOS M5000 Cell Imaging System, Life Technologies and are representative of the cell population. Scale bar: 100 µm. (**E**) The mean percentage of stained cells and the SD are represented in the graph. Moreover, the difference between the stained cells of the treated and non-treated cells was evaluated employing a one-way analysis of variance (ANOVA); **** *p* < 0.0001 significance. The experiment was carried out three times and each time five different images were analyzed in the EVOS m5000 Cell Imaging System, Life Technologies.

**Figure 4 marinedrugs-21-00224-f004:**
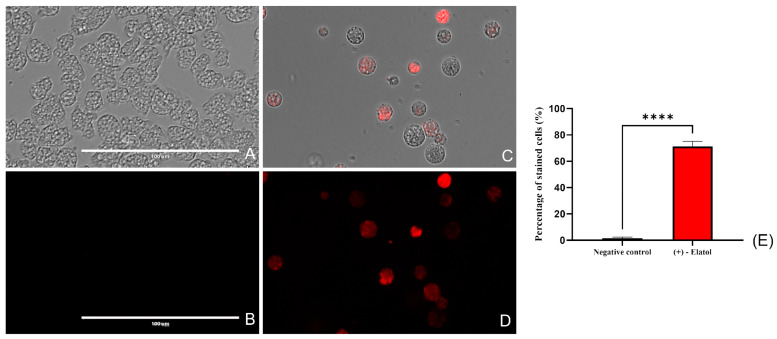
Trophozoites treated with the IC_90_ of (+)-elatol during 24h (**C**,**D**) show an intense red fluorescence due to the increase of the intracellular ROS generation. Cells incubated with bactocasitone (negative control) (**A**,**B**). Cells were exposed to CellROX^®^ Deep Red (5 μM, 30 min) at 37 °C in the dark. (**E**) The bar graph includes the mean value and the SD of the stained cells after performing the experiment in triplicate. Scale bar: 100 µm. The analysis of variance was performed in the GraphPad.PRISM^®^ 9.0 software and was determined by a one-way analysis of variance (ANOVA); **** *p* < 0.0001 significant differences when comparing treated cells to negative control. The number of stained cells was determined in the EVOS™ M5000 Software (Invitrogen by Thermo Fisher Scientific) processing five different images (×40) each time.

**Figure 5 marinedrugs-21-00224-f005:**
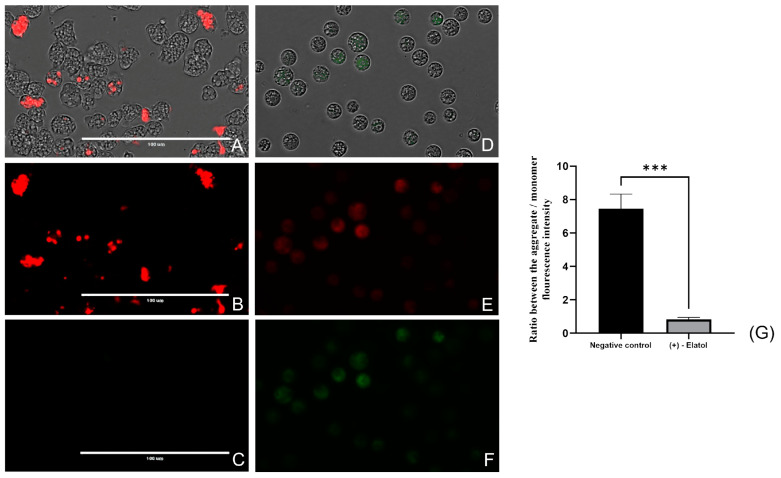
Evaluation of the mitochondrial membrane depolarization with the JC-1 stain (Cayman Chemical). Trophozoites treated with the IC_90_ of (+)-elatol (**1**) (**D**–**F**) and non-treated cells (**A**–**C**). The JC-1 dye emits red fluorescence in healthy conditions (negative control) (**A**,**B**) while the green fluorescence cannot be observed (**C**). However, the treatment of the cells with (+)-elatol (**1**) induces the depolarization of the mitochondrial membrane potential, showing a decrease in the intensity of the red fluorescence (as there are less J-aggregates) (**E**) and increasing the green fluorescence (a bigger amount of JC-1 monomers) (**F**). Images (×40) are representative of the cell population observed in the EVOS M5000 Cell Imaging System, Life Technologies, Spain. Scale bar: 100 µm. (**G**) Bar graph includes the ratio between the mean fluorescence emitted by the aggregate form of the JC-1 dye (red) and mean fluorescence emitted by the monomer form (green) of both treated and non-treated cells. Differences between the values were assessed using a one-way analysis of variance (ANOVA). Data are presented as means ± SD (N = 3); *** *p* < 0.001; significance differences when comparing treated cells to negative control. Mean fluorescence intensity values were calculated with the EVOS™ M5000 Software (Invitrogen by Thermo Fisher Scientific) tools. The experiments were conducted in triplicate and each time five different images were processed.

**Figure 6 marinedrugs-21-00224-f006:**
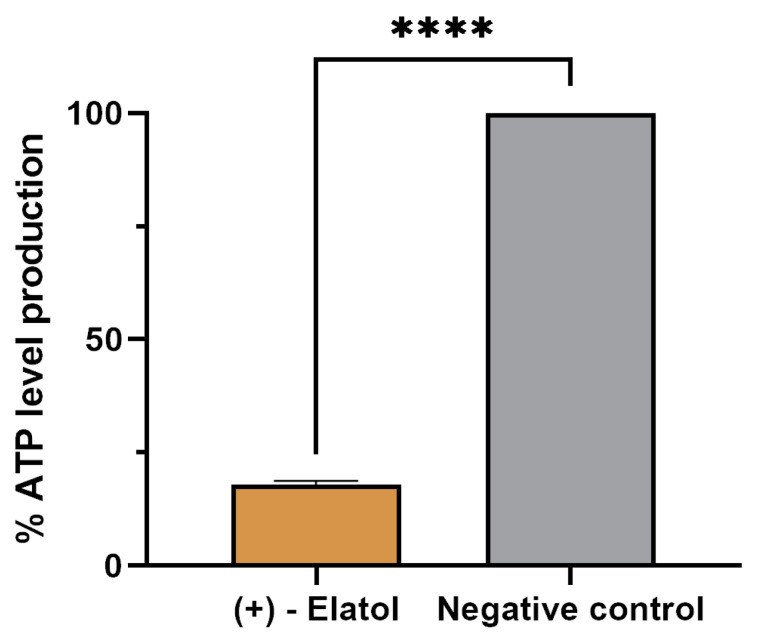
Effect of (+)-elatol (**1**) on ATP production. The results represent the percentage of ATP levels production relative to the negative control (untreated cells). Data represent the mean and the SD of three different assays. The ATP production is reduced in 82.11% in treated amoebae in comparison with the negative control. A one-way analysis of variance (ANOVA) in the GraphPad.PRISM^®^ 9.0 software was performed in order to analyze the difference between values; **** *p* < 0.0001 significant differences when comparing treated cells to negative control.

**Figure 7 marinedrugs-21-00224-f007:**
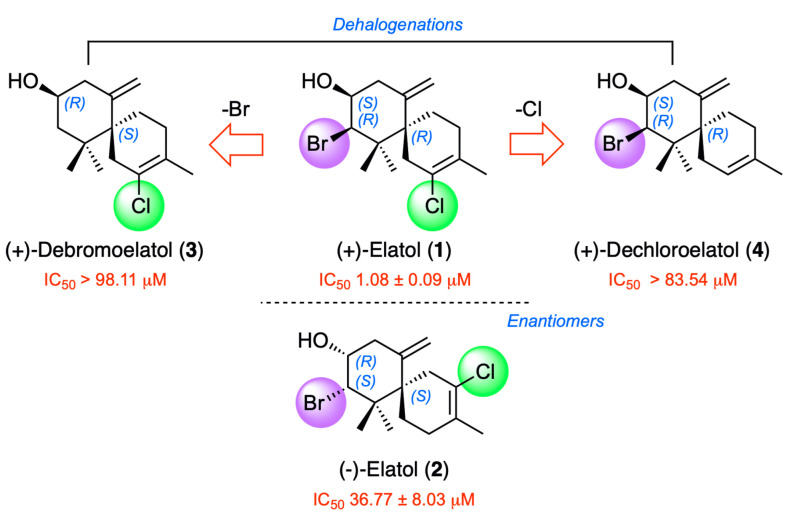
Structure–activity relationship analysis of elatol-related compounds **1**–**4**.

**Figure 8 marinedrugs-21-00224-f008:**
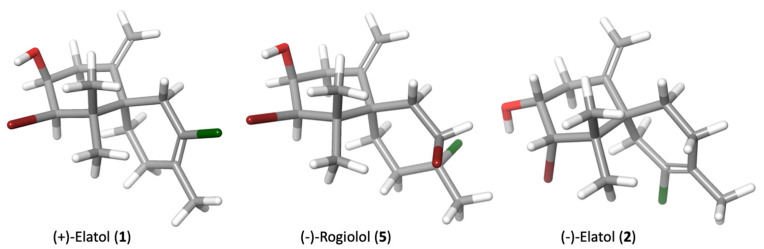
Minimized structures of sesquiterpenes 1, 2 and 5. Data calculated with Schrödinger Release 2022–2: Maestro Version 13.0, Schrödinger, LLC, New York, NY, 2021.

**Table 1 marinedrugs-21-00224-t001:** 50% inhibitory concentrations (IC_50_, μM) of the compounds isolated from the *L. dendroidea* algae against *N. fowleri* ATCC 30808™ and ^®^ 30215™ type strains and 50% cytotoxic concentration (CC_50_, μM) against murine macrophages.

Compound	*N. fowleri* ATCC 30808™	*N. fowleri* ATCC 30215™	Murine Macrophages J774.A1
(+)-Elatol (**1**)	1.08 ± 0.09	1.14 ± 0.09	61.52 ± 12.97
(−)-Elatol (**2**)	36.77 ± 8.03	38.03 ± 7.61	>150
(+)-Debromoelatol (**3**)	>98.11	-	-
(+)-Dechloroelatol (**4**)	>83.54	-	-
(−)-Rogiolol (**5**)	22.50 ± 5.81	9.71 ± 0.82	142.42 ± 17.19
(−)-3,10-Dibromo-4-chloro-alpha-chamigrane (**6**)	>62.71	-	-
(+)-Dendroidiol (**7**)	>71.08	-	-
(−)-Cartilagineol (**8**)	>60.30	-	-
(+)-Obtusol (**9**)	>60.30	-	-
(+)-Obtusane (**10**)	>62.71	-	-
Amphotericin B *	0.12 ± 0.03	0.17 ± 0,03	> 200
Miltefosine *	38.74 ± 4.23	81.57 ± 7.23	127.88 ± 8.85

* Amphotericin B and miltefosine: reference drugs.

## Data Availability

Data is contained within the article and Appendix A.

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
