# Peer review of "Chamigrane-Type Sesquiterpenes from Laurencia dendroidea as Lead Compounds against Naegleria fowleri"

_marinedrugs, 2023, doi:10.3390/md21040224_

Round 1

Reviewer 1 Report

The manuscript by  Iñigo Arberas-Jiménez et al, described the  (+)-Elatol was the most active compound against Naegleria fowleri trophozoites 27 with IC50 values of 1.08 M against the ATCC 30808™ strain and 1.14 M the ATCC 30215™ strain. 28 Furthermore, the activity of the (+)-elatol against the resistant stage of N. fowleri was also assessed, 29 showing great cysticidal properties with a very similar IC50 value (1.14 µM) to the one obtained for 30 the trophozoite stage. Its enantiomer (-)-elatol showed to be 34-fold less potent with IC50 of 36.77 31 M and 38.03 M. Structure-activity relationship analysis suggests that dehalogenation leads to a 32 significant decrease of activity. The lipophilic character of these compounds confers them as good 33 chemical scaffolds to develop new drugs to cross the brain-blood barrier.     the manuscript can be accepted  Marine Drugs journal after a major revision as following  (1) In the abstract author should include the compound numbers along with their names. (2) These compounds are new or already reported if not reported author should include the  1H NMR and 13 CNMR with their data with HRMS. (3) while checking the 1HNMR and 13 CNMR I found that the solvent used CDCl3 but in the manuscript, they mentioned that they used CD2Cl2. the author should check carefully with their spectra in the supporting information as well as in the manuscript. (3) they also reported the optical rotation of these compounds, the author should use the standard protocol for these rotations symbols.

Author Response

Response to reviewer 1

In the abstract author should include the compound numbers along with their names.

Answer: The compound numbers were included in the abstract along with their names, as proposed by the reviewer.

These compounds are new or already reported if not reported author should include the  1H NMR and 13 CNMR with their data with HRMS.

Answer: All compounds have been previously reported by the authors of this manuscript (References 39 and 40 before the first revision). The references have been included in the text in the results section.

  1. Machado, F.L.; Duarte, H.M.; Gestinari, L.M.; Cassano, V.; Kaiser, C.R.; Soares, A.R. Geographic Distribution of Natural Products Produced by the Red Alga Laurencia dendroidea J. Agardh. Chem Biodivers. 2016, 13, 845-51.
  2. Nocchi N.; Soares A.R.; Souto M.L.; Fernández J.J.; Martin M.N.; Pereira R.C. Detection of a chemical cue from the host seaweed Laurencia dendroidea by the associated mollusc Aplysia brasiliana. PLoS One. 2017, 12, e0187126.

While checking the 1HNMR and 13 CNMR I found that the solvent used CDCl3 but in the manuscript, they mentioned that they used CD2Cl2. the author should check carefully with their spectra in the supporting information as well as in the manuscript. (3) they also reported the optical rotation of these compounds, the author should use the standard protocol for these rotations symbols.

Answer: The solvent used was deuterated chloroform (CDCl3). This information has been corrected in the manuscript and the supporting information. The symbol of optical rotation has been corrected.

Reviewer 2 Report

The authors assessed chamigrane-type sesquiterpenes derived from Laurencia dendroidea with the aim to identify novel compounds with anti-Naegleria activities.

 The introduction is very informative. The methods are described in detail. The results are also described clearly. One of 10 test compounds isolated and characterized by NMR/MS was quite promising against both trophozoite and cyst forms of Naegleria fowleri strains. The tables and figures are very helpful for clear understanding of the work.

 English needs many minor corrections, especially the unnecessary use of “the” in many sentences, and spelling/typographical errors.

 MAJOR COMMENTS:

 The abstract is focused on in vitro activity of elatol. Results on cytotoxicity and programmed cell death (plasma membrane permeability, ROS, mitochondrial function, ATP production) are not mentioned. These results should be added (in 1 or 2 sentences) to the abstract.

 Also, the text in the concluding paragraph (section 5. Conclusions) is quite similar to the text in the abstract. I think that, instead of repeating what is in the abstract, the authors should add a short new paragraph at the end of the Discussion section on (i) the limitations of the study and (ii) future directions or perspectives for further development of elatol or other compounds derived from Laurencia spp.   

 MINOR COMMENTS:

 ABSTRACT:

 Line 31: was shown to be 34-fold less potent with IC50 of 36.7 µM and 38.0 µM against the two strains, respectively

 Line 32: Analysis of structure-activity relationship

 Line 33: “lipophilic character of these compounds confers them as good chemical scaffolds to develop new drugs to cross the blood brain barrier” [not “brain blood barrier”]: This sentence should be rewritten correctly. What (which atoms, molecules) in the chemical scafford confer lipophilic character to new drugs that is favorable for crossing the blood brain barrier?

 INTRODUCTION:

 Line 44: capable of infecting humans

 Line 53: by ingestion…when immersed in water

 Line 56: G-protein-coupled receptors (GPCR)

 Lines 62-65: These signs and symptoms usually appear…” “A few days after the onset of these first signs and symptoms…” [my comment: headache is a symptom, not a clinical sign]; the intracranial and cerebrospinal fluid pressure increaseS [singular verb]…

 Line 66: lumbar puncture

 Line 85: The natural and marine environment [singular noun] is… OR …environments are…

 Lines 86-87: …metabolite that offers… OR …metabolites that offer…  to design antiprotozoal drugs

 Line 92: …as [delete “one”] a prolific producer

 Line 99: Chamigrene is… OR chamigrenes are…

 Line 105: antiparasitic (spelling)

 Line 108: This figure caption should be placed before line 105?

 RESULTS:

 General comments for the results section: Many of the sentences in the Results section should be in the past tense. For example, in line 214, the ratio…showed a significant difference…

 Line 113: Laurencia dendroidea [in italics]

 Line 118: mass spectrometry (MS)

 Line 120: CHCl3 (“3” in subscript)

 Line 121: These data confirm that compound 1 [delete “it”] is either the dextrorotary/(+) or levorotary/(-) enantiomer of compound 2 ? Please clarify.

 Line 125: against both N. fowleri [in italics] ATCC 30808 and ATCC 30215 strains (spelling)

 Lines 126-127: was active at similar concentrations as amphotericin B

 Lines 130-132: 50% inhibitory concentrations (IC50, µM); L. obtuse [italics]; N. flowleri [italics]; 50% cytotoxic concentration (CC50, µM). Please separate the legend (reference drugs amphotericin B and miltefosine [spelling]) from table title. The legend (caption) should be under the table.

 Line 135: 1.14 ± 0.09 µM

 Line 144: characteristic of PCD [delete “programmed cell death”]

 Line 149: during PCD [delete “programmed cell death”]

 Line 151: S13 show [delete the comma] an intense fluorescence

 Line 153: corresponds to PI [delete “propidium iodide”]

 Line 157: chromatin (spelling); emits (spelling)

 Line 161: cell population and (spelling)

 Lines 162-165: bar graph; standard deviation (SD); ns = not significant

 Lines 169-170: plasma membrane

 Line 171: significantly higher

 Line 177: Evaluation of plasma membrane integrity

 Line 179: impermeable to healthy [spelling] (untreated) cells in which fluorescence is not observed

 Line 181: emitting [spelling]

 Lines 184-185: the difference…non-treated cells was evaluated

 Line 202: significant differences

 Line 203: The number of stained cells was determined OR The number of stained cells was assessed…processing…

 Line 219: JC-1 dye emits

 Line 221: showing a decrease

 Line 225: bar graph includes the ratio

 Line 231: five different images were processed

 Line 233: in 82.1% of treated cells compared to untreated cells

 Line 238: 82.1% in treated amoeba

 Line 241: significant differences

 DISCUSSION:

 Line 253: was shown to be; IC50 [subscript]

 Lines 254-255: loss of any halogen atom, bromine…leads

 Lines 255-256: IC50 [subscript]

 Lines 256-270: Please delete the text in these paragraphs: “The Materials and Methods should be described with sufficient details… can be briefly described and appropriate cited.” “Research manuscripts reporting large datasets…They must be provided prior to publication.” “Interventionary studies…ethical approval code.”

 Line 275: IC50 [subscript]

 Lines 279-280: The figure title and caption should be in line 273?

 Line 286: plasma membrane

 Line 291: poured into the intercellular space OR released/discharged into the intercellular space

 Line 292: side effects. [Period, start a new sentence] However, through…

 Line 297: Place [36] at the end of the sentence

 METHODS:

 Line 312: Evaluation of programmed cell death (PCD) induction

 Line 318: CH2Cl2 [“2” in subscript]

 Line 320: high-performance liquid chromatography (HPLC)

 Line 324: nuclear magnetic resonance (NMR)

 Line 328: deuterochloroform (CDCl3)

 Line 333: Naegleria fowleri [in italics]

 Line 337: Universidad de La Laguna, Spain

 Line 340: fetal bovine serum (FBS)

 Line 342: Naegleria fowleri [in italics] ATCC 30808 strain [delete “s”] was used

 Line 351: 50% and 90% inhibitory concentrations (IC50 and IC90)

 50% cytotoxic concentration (CC50)

 Line 359: The in vitro activity of the compounds against Naegleria fowleri [in italics] cyst was determined…

 Line 362: the media were removed OR the medium was removed

 Line 363: incubated for additional 72 hours

 Line 365: in section 4.4

 Line 368: Naegleria fowleri [in italics]

 Line 380: healthy cells will not show any red fluorescence…

 Lines 381-384: plasma membrane

 Line 389: Naegleria fowleri [in italics]

 Line 392: reactive oxygen species (ROS)

 Line 405: Promega Biotech Ibérica [in small letters]

 Line 407: I think that the authors should add another section here: 4.8: Statistical analysis. The statistical tests described in several figure captions (for example, Fig. 2 lines 163-165; Fig. 3 lines 184-186; Fig. 4 lines 200-202; Fig. 5 lines 227-229; Fig. 6 lines 239-241), as well as the statistical software, for example lines 279-280 Schrödinger, should be described in the Methods section under “statistical analysis.”

 CONCLUSIONS:

 Line 417: structure-activity relationship (SAR)

 REFERENCES:

 References: Please use the same format for all references listed. In the article titles, the first letters of each word should be in small letters, except for the first word and proper names: see Ref 1, 7, 8, 11, 13, 15 and others.

 Ref 2: Please check the authors and journal name of this reference. I found: Peralta Rodríguez ML, Ayala Oviedo, JJ. Amibas de vida libre en seres humanos. Revista Salud Uninorte 2009, 25(2), 280-292.

 Ref 5: Please check the publication year. It should be 2020?: Stahl LM, Olson JB. Environmental abiotic and biotic factors affecting the distribution and abundance of Naegleria fowleri. FEMS Microbiol Ecol. 2020, 97(1), fiaa238.

 Ref 7: The author names (first name/family name) should be checked. I found: Grace E, Asbill S, Virga K. Naegleria fowleri: pathogenesis, diagnosis, and treatment options. Antimicrob Agents Chemother. 2015, 59(11), 6677-81.

 Ref 15: This reference is not complete. Please add the article number: Iqbal K, Abdalla SAO, Anwar A, Iqbal KM, Shah MR, Anwar A, Siddiqui R, Khan NA. Isoniazid conjugated magnetic nanoparticles loaded with amphotericin B as a potent antiamoebic agent against Acanthamoeba castellanii. Antibiotics (Basel). 2020, 9(5), 276.

 Ref 17: Please check the author names: Defillo A, Nussbaum PE, Hariharan P, et al. Hypothermia as an adjunctive treatment in pediatric patients with Naegleria fowleri: a systematic review. J Pediatr Neurol Neurosci 2021, 5(1), 105-110.

 Ref 28: Please check the article number (instead of the temporary page numbers “1-13”): Arberas-Jiménez I, García-Davis S, Rizo-Liendo A, Sifaoui I, Reyes-Batlle M, Chiboub O, Rodríguez-Expósito RL, Díaz-Marrero AR, Piñero JE, Fernández JJ, Lorenzo-Morales J. Laurinterol from Laurencia johnstonii eliminates Naegleria fowleri triggering PCD by inhibition of ATPases. Sci Rep. 2020, 10(1), 17731.

 Ref 29: The article number is missing: Arberas-Jiménez I, García-Davis S, Rizo-Liendo A, Sifaoui I, Morales EQ, Piñero JE, Lorenzo-Morales J, Díaz-Marrero AR, Fernández JJ. Cyclolauranes as plausible chemical scaffold against Naegleria fowleri. Biomed Pharmacother. 2022, 149, 112816.

 Ref 30: Please complete this reference: Massarani S. Phytochemical and biological properties of sesquiterpene constituents from the marine red seaweed Laurencia: a review. Nat Prod Chem Res 2014, 2(5), 147.

 Ref 36: Please complete this reference (volume, article number): Petraitis V, Petraitiene R, Valdez JM, Pyrgos V, Lizak MJ, Klaunberg BA, Kalasauskas D, Basevicius A, Bacher JD, Benjamin DK Jr, Hope WW, Walsh TJ. Amphotericin B penetrates into the central nervous system through focal disruption of the blood brain barrier in experimental hematogenous Candida meningoencephalitis. Antimicrob Agents Chemother. 2019, 63(12), e01626-19.

 Ref 41: Please correct the article number (and delete the temporary page numbers “1-5”): Arberas-Jiménez I, Rizo-Liendo A, Sifaoui I, Chao-Pellicer J, Piñero JE, Lorenzo-Morales J. A fluorometric assay for the in vitro evaluation of activity against Naegleria fowleri cysts. Microbiol Spectr. 2022, 10(4), e0051522.

Author Response

Response to reviewer 2

Major revision:

The abstract is focused on in vitro activity of elatol. Results on cytotoxicity and programmed cell death (plasma membrane permeability, ROS, mitochondrial function, ATP production) are not mentioned. These results should be added (in 1 or 2 sentences) to the abstract.

A sentence in the abstract has been added mentioning the low cytotoxicity of (+)-elatol and the induction of the different cellular events when treating the amoebae. “Moreover, at low concentrations (+)-elatol (1) showed no toxic effect towards murine macrophages and could induced the appearance of different cellular events related to the programmed cell death, such as increase of the plasma membrane permeability, reactive oxygen species overproduction, mitochondrial malfunction or chromatin condensation”.

Also, the text in the concluding paragraph (section 5. Conclusions) is quite similar to the text in the abstract. I think that, instead of repeating what is in the abstract, the authors should add a short new paragraph at the end of the Discussion section on (i) the limitations of the study and (ii) future directions or perspectives for further development of elatol or other compounds derived from Laurencia spp.  

A new paragraph has been added at the end of the discussion section regarding the limitations and future direction perspectives. “This work represents a first approach to elucidate the in vitro activity and the mechanism of action of the (+)-elatol against N. fowleri using fluorescence microscopy. A proteomic study could deepen and confirming the effect of the bioactive molecule in the present protozoa.”

MINOR COMMENTS:

ABSTRACT:

Line 31: was shown to be 34-fold less potent with IC50 of 36.7 µM and 38.0 µM against the two strains, respectively.

Corrected as suggested.

Line 32: Analysis of structure-activity relationship

Modified as suggested.

Line 33: “lipophilic character of these compounds confers them as good chemical scaffolds to develop new drugs to cross the blood brain barrier” [not “brain blood barrier”]: This sentence should be rewritten correctly. What (which atoms, molecules) in the chemical scaffold confer lipophilic character to new drugs that is favorable for crossing the blood brain barrier?

The sentence has been rewritten. The chemical structure (presence of halogens and carbon skeleton) of these compounds show high lipophilicity with calculated log P values higher than 3.65 (elatol). This is an essential property for substances to cross the blood brain barrier, therefore they represent interesting chemical scaffolds to develop new drugs.

INTRODUCTION:

Line 44: capable of infecting humans

Corrected as suggested.

Line 53: by ingestion…when immersed in water

Corrected as “Most infections begin by inhalation of contaminated water containing the amoeba in the trophozoite form”.

Line 56: G-protein-coupled receptors (GPCR)

Changed as suggested.

Lines 62-65: These signs and symptoms usually appear…” “A few days after the onset of these first signs and symptoms…” [my comment: headache is a symptom, not a clinical sign]; the intracranial and cerebrospinal fluid pressure increaseS [singular verb]…

Changed as These signs and symptoms usually appear during the first few days after infection and usually include fever, headache, vomiting, neck stiffness and convulsions. A few days after the onset of these first sign and symptoms, the intracranial and cerebrospinal fluid pressure increases preceding the death of the patient.

Line 66: lumbar puncture

Modified as suggested.

Line 85: The natural and marine environment [singular noun] is… OR …environments are…

Modified as “The natural and the marine environments are…”.

Lines 86-87: …metabolite that offers… OR …metabolites that offer…  to design antiprotozoal drugs

Changed as “…active secondary metabolites that offer the opportunity to design antiprotozoal drugs.”

Line 92: …as [delete “one”] a prolific producer

Corrected as suggested.

Line 99: Chamigrene is… OR chamigrenes are…

Modified as “The chamigrenes are characterized…”.

Line 105: antiparasitic (spelling)

Corrected.

Line 108: This figure caption should be placed before line 105?

The figure caption has been placed before the last paragraph.

RESULTS:

General comments for the results section: Many of the sentences in the Results section should be in the past tense. For example, in line 214, the ratio…showed a significant difference…

The results section has been revised and changed the sentence to write them in the past tense.

Line 113: Laurencia dendroidea [in italics]

Corrected.

Line 118: mass spectrometry (MS)

Changed as recommended.

Line 120: CHCl3 (“3” in subscript)

Corrected.

Line 121: These data confirm that compound 1 [delete “it”] is either the dextrorotary/(+) or levorotary/(-) enantiomer of compound 2 ? Please clarify.

Line 125: against both N. fowleri [in italics] ATCC 30808 and ATCC 30215 strains (spelling)

Corrected.

Lines 126-127: was active at similar concentrations as amphotericin B

Changed as suggested.

Lines 130-132: 50% inhibitory concentrations (IC50, µM); L. obtuse [italics]; N. flowleri [italics]; 50% cytotoxic concentration (CC50, µM). Please separate the legend (reference drugs amphotericin B and miltefosine [spelling]) from table title. The legend (caption) should be under the table.

Corrected as suggested. The legend “reference drugs amphotericin B and miltefosina” has been separated from the table title and placed under the table.

Line 135: 1.14 ± 0.09 µM

Corrected.

Line 144: characteristic of PCD [delete “programmed cell death”]

Corrected.

Line 149: during PCD [delete “programmed cell death”]

Corrected.

Line 151: S13 show [delete the comma] an intense fluorescence

Corrected.

Line 153: corresponds to PI [delete “propidium iodide”]

Modified as suggested.

Line 157: chromatin (spelling); emits (spelling)

Corrected as suggested.

Line 161: cell population and (spelling)

Modified as suggested.

Lines 162-165: bar graph; standard deviation (SD); ns = not significant

Changed as suggested.

Lines 169-170: plasma membrane

Modified.

Line 171: significantly higher

Corrected

Line 177: Evaluation of plasma membrane integrity

Changed as suggested.

Line 179: impermeable to healthy [spelling] (untreated) cells in which fluorescence is not observed.

Changed as proposed.

Line 181: emitting [spelling]

Spelling corrected.

Lines 184-185: the difference…non-treated cells was evaluated

Corrected as suggested.

Line 202: significant differences

Modified as proposed.

Line 203: The number of stained cells was determined OR The number of stained cells was assessed…processing…

Changed as suggested.

Line 219: JC-1 dye emits.

Changed as suggested.

Line 221: showing a decrease.

Modified as proposed.

Line 225: bar graph includes the ratio.

Corrected

Line 231: five different images were processed.

Revised.

Line 233: in 82.1% of treated cells compared to untreated cells.

Changed as suggested.

Line 238: 82.1% in treated amoeba.

Modified as suggested.

Line 241: significant differences.

Corrected as suggested.

DISCUSSION:

Line 253: was shown to be; IC50 [subscript].

Changed as recommended.

Lines 254-255: loss of any halogen atom, bromine…leads

Corrected

Lines 255-256: IC50 [subscript]

Corrected.

Lines 256-270: Please delete the text in these paragraphs: “The Materials and Methods should be described with sufficient details… can be briefly described and appropriate cited.” “Research manuscripts reporting large datasets…They must be provided prior to publication.” “Interventionary studies…ethical approval code.”

Corrected as suggested.

Line 275: IC50 [subscript]

Revised.

Lines 279-280: The figure title and caption should be in line 273?

Corrected as requested.

Line 286: plasma membrane

Corrected.

Line 291: poured into the intercellular space OR released/discharged into the intercellular space.

Corrected as “cytosolic constituents are poured into the intercellular space”

Line 292: side effects. [Period, start a new sentence] However, through…

Modified as suggested.

Line 297: Place [36] at the end of the sentence.

Changed as proposed.

METHODS:

Line 312: Evaluation of programmed cell death (PCD) induction.

The title of the section 4.2 has been deleted, it was incorrectly located.

Line 318: CH2Cl2 [“2” in subscript].

Modified as suggested.

Line 320: high-performance liquid chromatography (HPLC)

Modified as suggested.

Line 324: nuclear magnetic resonance (NMR)

Changed as proposed.

Line 328: deuterochloroform (CDCl3)

Changed as suggested.

Line 333: Naegleria fowleri [in italics]

Corrected.

Line 337: Universidad de La Laguna, Spain.

Corrected

Line 340: fetal bovine serum (FBS).

Changed as suggested.

Line 342: Naegleria fowleri [in italics] ATCC 30808 strain [delete “s”] was used.

Corrected.

Line 351: 50% and 90% inhibitory concentrations (IC50 and IC90)

50% cytotoxic concentration (CC50)

Modified as recommended.

Line 359: The in vitro activity of the compounds against Naegleria fowleri [in italics] cyst was determined…

Corrected as suggested.

Line 362: the media were removed OR the medium was removed.

Changed as “…the medium was removed…”

Line 363: incubated for additional 72 hours.

Changed as proposed.

Line 365: in section 4.4.

Changed as “4.3” since the section “4.2 Programmed cell death (PCD) induction evaluation” was deleted and the numbers of the sections have changed.

Line 368: Naegleria fowleri [in italics].

Corrected.

Line 380: healthy cells will not show any red fluorescence…

Changed as suggested.

Lines 381-384: plasma membrane

Corrected.

Line 389: Naegleria fowleri [in italics]

Corrected.

Line 392: reactive oxygen species (ROS)

Added as recommended.

Line 405: Promega Biotech Ibérica [in small letters]

Changed as proposed

Line 407: I think that the authors should add another section here: 4.8: Statistical analysis. The statistical tests described in several figure captions (for example, Fig. 2 lines 163-165; Fig. 3 lines 184-186; Fig. 4 lines 200-202; Fig. 5 lines 227-229; Fig. 6 lines 239-241), as well as the statistical software, for example lines 279-280 Schrödinger, should be described in the Methods section under “statistical analysis.”

A 4.7 section titled “Statistical analysis” has been added where all the details related to the statistical issues are explained.

Maestro Software (Schrödinger) Version 13.0, was used to obtain minimized structures of Figure 8, and therefore it should not be included in the statistical analysis section.

CONCLUSIONS:

Line 417: structure-activity relationship (SAR).

Changed as proposed.

REFERENCES

References: Please use the same format for all references listed. In the article titles, the first letters of each word should be in small letters, except for the first word and proper names: see Ref 1, 7, 8, 11, 13, 15 and others.

All the reference list has ben revised and the title of the articles corrected.

Ref 2: Please check the authors and journal name of this reference. I found: Peralta Rodríguez ML, Ayala Oviedo, JJ. Amibas de vida libre en seres humanos. Revista Salud Uninorte 2009, 25(2), 280-292.

Corrected.

Ref 5: Please check the publication year. It should be 2020?: Stahl LM, Olson JB. Environmental abiotic and biotic factors affecting the distribution and abundance of Naegleria fowleri. FEMS Microbiol Ecol. 2020, 97(1), fiaa238.

Revised and changed the publication year.

Ref 7: The author names (first name/family name) should be checked. I found: Grace E, Asbill S, Virga K. Naegleria fowleri: pathogenesis, diagnosis, and treatment options. Antimicrob Agents Chemother. 2015, 59(11), 6677-81.

The author names have been corrected.

Ref 15: This reference is not complete. Please add the article number: Iqbal K, Abdalla SAO, Anwar A, Iqbal KM, Shah MR, Anwar A, Siddiqui R, Khan NA. Isoniazid conjugated magnetic nanoparticles loaded with amphotericin B as a potent antiamoebic agent against Acanthamoeba castellanii. Antibiotics (Basel). 2020, 9(5), 276.

The reference has been completed.

Ref 17: Please check the author names: Defillo A, Nussbaum PE, Hariharan P, et al. Hypothermia as an adjunctive treatment in pediatric patients with Naegleria fowleri: a systematic review. J Pediatr Neurol Neurosci 2021, 5(1), 105-110.

The author names have been changed.

Ref 28: Please check the article number (instead of the temporary page numbers “1-13”): Arberas-Jiménez I, García-Davis S, Rizo-Liendo A, Sifaoui I, Reyes-Batlle M, Chiboub O, Rodríguez-Expósito RL, Díaz-Marrero AR, Piñero JE, Fernández JJ, Lorenzo-Morales J. Laurinterol from Laurencia johnstonii eliminates Naegleria fowleri triggering PCD by inhibition of ATPases. Sci Rep. 2020, 10(1), 17731.

The article number has been updated.

Ref 29: The article number is missing: Arberas-Jiménez I, García-Davis S, Rizo-Liendo A, Sifaoui I, Morales EQ, Piñero JE, Lorenzo-Morales J, Díaz-Marrero AR, Fernández JJ. Cyclolauranes as plausible chemical scaffold against Naegleria fowleri. Biomed Pharmacother. 2022, 149, 112816.

The article number has been added.

Ref 30: Please complete this reference: Massarani S. Phytochemical and biological properties of sesquiterpene constituents from the marine red seaweed Laurencia: a review. Nat Prod Chem Res 2014, 2(5), 147.

The reference has been completed.

Ref 36: Please complete this reference (volume, article number): Petraitis V, Petraitiene R, Valdez JM, Pyrgos V, Lizak MJ, Klaunberg BA, Kalasauskas D, Basevicius A, Bacher JD, Benjamin DK Jr, Hope WW, Walsh TJ. Amphotericin B penetrates into the central nervous system through focal disruption of the blood brain barrier in experimental hematogenous Candida meningoencephalitis. Antimicrob Agents Chemother. 2019, 63(12), e01626-19.

The reference has been completed.

Ref 41: Please correct the article number (and delete the temporary page numbers “1-5”): Arberas-Jiménez I, Rizo-Liendo A, Sifaoui I, Chao-Pellicer J, Piñero JE, Lorenzo-Morales J. A fluorometric assay for the in vitro evaluation of activity against Naegleria fowleri cysts. Microbiol Spectr. 2022, 10(4), e0051522.

The reference has been corrected.

Reviewer 3 Report

The ms entitled "Chamigrane-type sesquiterpenes from Laurencia dendroidea as lead compounds against Naegleria fowleri" presented by Arberas-Jiménez et al.  is well structured and the objectives are clear. The Material and methods section is adequate. In general, the results support the conclusions made by the authors. My minor comments can be found below:

- I recommend the revision of English language throughout the text, especially regarding the excessive use of the article "the".

- Figure captions are not placed correctly. 

- Check italics throughout the text.

- L. 125 - "strains".

- L. 126 - Define "similar", perhaps using values.

- L. 132 - "miltefosine".

- L. 143 - Rewrite sentence.

- L. 197 - "increase".

- L.256-269: delete.

- L.367 - For the determination of the type of cell death that induces the (+)-elatol (1) different 367 commercial kits were used. 

Author Response

Reviewer 3

- I recommend the revision of English language throughout the text, especially regarding the excessive use of the article "the".

The English has been revised.

- Figure captions are not placed correctly.

The figure captions have been corrected.

- Check italics throughout the text.

The text has been revised and the italics corrected.

- L. 125 - "strains".

Corrected.

- L. 126 - Define "similar", perhaps using values.

The sentence has been changed to “Additionally, (+)-elatol was active at similar concentration values as amphotericin B”.

- L. 132 - "miltefosine".

Corrected.

- L. 143 - Rewrite sentence.

The sentence has been rewritten as follows: “The assesment of the type of cell death induced by (+)-elatol was conducted using several techiques to highlight different cellular events considered as PCD markers”.

- L. 197 - "increase".

Corrected.

- L.256-269: delete.

Corrected as suggested.

- L.367 - For the determination of the type of cell death that induces the (+)-elatol (1) different 367 commercial kits were used.

In the text it is written like “For the determination of the type of cell death that induces the (+)-elatol (1) different commercial kits were used”. The 367 may be due to an informatic error.

Round 2

Reviewer 1 Report

The author carried a substantial  revision and I am satisfy but sill the issue must be resolved before it publish. In supplementary material still they did not corrected CD2Cl2 to CDCl3